# Effect of the Plasma Gas Type on the Surface Characteristics of 3Y-TZP Ceramic

**DOI:** 10.3390/ijms23063007

**Published:** 2022-03-10

**Authors:** Sung-Un Kang, Chul-Ho Kim, Hee-Kyung Kim, Ye-Won Yoon, Yu-Kwon Kim, Seung-Joo Kim

**Affiliations:** 1Department of Otolaryngology, Ajou University School of Medicine, Suwon 16499, Korea; cows79@ajou.ac.kr; 2Department of Molecular Science and Technology, Ajou University School of Medicine, Suwon 16499, Korea; 3Department of Prosthodontics, Institute of Oral Health Science, Ajou University School of Medicine, Suwon 16499, Korea; 4Department of Chemistry and Department of Energy Systems Research, Ajou University, Suwon 16499, Korea; yyw0902@ajou.ac.kr (Y.-W.Y.); yukwonkim@ajou.ac.kr (Y.-K.K.); sjookim@ajou.ac.kr (S.-J.K.)

**Keywords:** plasma gases, zirconium oxide, surface properties, nitrogen, wettability

## Abstract

Plasma surface treatment can be an attractive strategy for modifying the chemically inert nature of zirconia to improve its clinical performance. This study aimed to clarify the effect of plasma gas compositions on the physicochemical surface modifications of 3 mol% yttria-stabilized zirconia (3Y-TZP). The cold, atmospheric plasma discharges were carried out by using four different plasma gases, which are He/O_2_, N_2_/Ar, N_2_, and Ar from an application distance of 10 mm for 60 s. Static contact angles were measured to define the surface free energy. Changes in elemental composition, surface crystallinity, and surface topography were assessed with X-ray photoelectron spectroscopy (XPS), X-ray diffraction (XRD), confocal laser scanning microscopy (CLSM), and scanning electron microscopy (SEM), respectively. A significant decrease in water contact angle was observed in all plasma groups with the lowest value of 69° in the N_2_/Ar group. CLSM and SEM investigations exhibited no morphological changes in all plasma groups. XPS revealed that a reduction in the surface C content along with an increase in O content was pronounced in the case of N_2_/Ar compared to others, which was responsible for high hydrophilicity of the surface. XRD showed that the changes in crystallite size and microstrain due to oxygen atom displacements were observed in the N_2_/Ar group. The N_2_/Ar plasma treatment may contribute to enhancing the bioactivity as well as the bonding performance of 3Y-TZP by controlling the plasma-generated nitrogen functionalities.

## 1. Introduction

The surface modification strategies for biomaterials help to tailor the outcome of biological-material interactions by controlling the surface energy, biocompatibility, and adhesion strength of the substrates [1]. Plasma modification is one way to alter the surface properties of biomaterials with high-energy ion bombardment through physical collisions or chemical reactions of the excited gas molecules [2]. Atmospheric-pressure glow discharge (APGD) plasma, especially cold atmospheric plasma (CAP), has recently attracted a great deal of interest for a variety of industrial and medical applications, such as surface processing [3], film deposition [3], ozone production for water purification [4], biomedical decontamination [4], wound healing [5,6], muscle regeneration [7], and cancer treatment [8,9], etc. CAP, also named non-thermal plasma (NTP), consists of a partially ionized gas that is not in the thermodynamic equilibrium. CAP generates a large amount of chemically reactive oxygen and nitrogen species (RONS) in biological systems [3,10]. One of the typical CAP sources is dielectric-barrier discharge (DBD), which is responsible for a self-pulsing plasma operation with an insulating (dielectric) material in the discharge gap. A DBD system requires a high voltage AC source (1–100 kV_rms_) in the kHz range [11].

Zirconia ceramics stabilized with 3 mol% of yttria (3Y-TZP) have been widely used in dental applications for fabricating crown and bridge restorations, dental implants, orthodontic brackets, and endodontic posts due to their excellent biocompatibility, sufficient mechanical strength, and high esthetic potential [12]. However, zirconia is characterized by a chemically inert surface with low reactivity which limits a reliable bonding with resin cement, cell adhesion, or osseointegration [13,14]. Although mechanical surface treatments tended to increase the bond strength of zirconia with resin cement through micromechanical retention, the induced cracks and surface damages could cause a deterioration of fracture resistance of zirconia [15]. Therefore, the plasma surface treatment has been considered as an alternative to the mechanical surface treatment aiming to raise the surface energy of materials by the generation of polar groups at the surface [16]. Although the plasma surface treatments have been found to increase the surface hydrophilicity of 3Y-TZP without altering the surface topography [17], several studies demonstrated that the plasma treatment did not significantly enhance relevant shear bond strength (SBS) between zirconia and composite resin [18,19]. Conversely, oxygen radicals created in plasmas could mainly remove organic contaminants at the surface with a potential prospect for antibacterial efficacy around the zirconia abutments [10].

Nevertheless, the plasma performance of all those applications significantly depends on the experimental parameters. One of the most important characteristics is the gas species [20,21]. A noble gas, such as helium (He) or argon (Ar), as a carrier gas is usually employed to trigger CAP discharge because of its low breakdown voltage. However, a previous study reported that no chemical reaction was found on the micro-organisms with argon gas plasma due to the inert nature of argon despite intensive ion bombardments [22]. By contrast, some reactive gases, such as oxygen, nitrogen, or air can be mixed in small quantities into the background of noble gas for the production of chemically active species, such as O_3_, OH, H_2_O_2_, NO, and OH radicals at low temperatures with a reduction in the breakdown voltage [23,24]. In a plasma generated from a gas mixture, the excited noble gas can ionize the reactive gas by energy transfer (Penning ionization) via collision, resulting in a change in the discharge characteristics [25].

To enhance the reactive level of the zirconia surface during the plasma treatment, investigation of the electrical conductivity and quantitative ion concentration in terms of changes in surface electrochemistry should be conducted in an attempt to increase the bonding efficiency or osseointegration of zirconia implants. In a recent study, a new functional group was generated on the zirconia surface by carbon and nitrogen plasma ion implantation technology, resulting in the enhanced bioactivity and cytocompatibility of 3Y-TZP [14]. However, there has yet been no report on the plasma-zirconia interactions from the perspective of the role of gas compositions that are responsible for the kinetics of chemical reactions at the surface. Therefore, the present study highlights the effect of plasma gas compositions on the physicochemical surface modifications of 3Y-TZP to gain some insight into the plasma-generated ionic functionalities. In this study, He/O_2_ mixture, N_2_/Ar mixture, N_2_, and Ar were provided as feed gases for plasma generation. Characterization methods such as contact angle, X-ray photoelectron spectroscopy (XPS), X-ray diffraction (XRD) and Rietveld analysis, confocal laser scanning microscopy (CLSM), and scanning electron microscopy (SEM) were used to identify any changes in surface energy, surface chemistry, phase composition, and morphology of 3Y-TZP. The null hypothesis tested in this study was that there would be no difference in the physicochemical surface properties among 3Y-TZP ceramics treated with various gas plasmas.

## 2. Results

### 2.1. Surface Energy Changes by Irradiation of Various Gas Plasmas

Figure 1 shows the contact angles with sessile drop images (A) and the values of γ^total^, γ^d^, and γ^p^ (B) in the zirconia specimens subjected to each plasma treatment. Table 1 summarizes contact angles measured on the specimens of each plasma group. A significant decrease in water contact angle was observed after plasma exposure in all plasma groups with the lowest value of 69° in the N_2_/Ar group. The diiodomethane contact angles remained roughly constant in all plasma groups except in the Ar group, presented in Figure 1A and Table 1. Total surface energy significantly increased after the plasma treatment in all plasma groups and these results were mainly consistent with an increase in γ^p^. The largest value of γ^p^ was measured in the N_2_/Ar group (Figure 1B).

### 2.2. Surface Chemistry Changes by Irradiation of Various Gas Plasmas

Figure 2 shows the XPS C 1s, O 1s, N 1s, Y 3d, and Zr 3d core-level spectra, and Figure 3 shows the atomic percentages (at%) of these elements determined by XPS and the carbon/oxygen ratio in all groups. The nitrogen content on the zirconia surface increased after the plasma treatment in all plasma groups, but only to a few percent ranges (1–2%) as illustrated in Figure 3A,B. An increase in oxygen content and a decrease in carbon content were found in both N_2_/Ar and He/O_2_ groups, with the largest decrease in C content and the largest increase in O content in the N_2_/Ar group (Figure 3A). As can be seen in Figure 3C, the lowest value of C/O ratio was observed in the N_2_/Ar group. This would be related to the largest increase in the surface hydrophilicity in the N_2_/Ar group, possibly by the production of a high level of oxygen-based radicals [26].

The N 1s photoelectron region, Figure 2B, showed a new component at a binding energy of 406.5 eV in the N_2_/Ar group, which was associated with the presence of nitrate (NO_3_^−^) species [27]. This was due to the adsorption of a nitrogen atom (N) on the zirconia surface yielding N-containing functionalities [28]. A second component that appeared at a binding energy of around 399.1 eV was associated with the typical bonding states of N in zirconium oxynitride, or ZrO_x_N_y_ which confirmed the introduction of nitrogen in the zirconia lattice [29,30]. The resulting N-doped zirconium oxynitride products were found in all tested groups without a direct N_2_ feed, exhibiting the highest intensity in the N_2_/Ar group (Figure 2C and Figure 3B). This can be attributed to the fact that the surrounding air might have been involved in the interactions with zirconia surfaces during the plasma treatments [10].

Carbon content on the zirconia surface subjected to each plasma gas was shown in Figure 2A. The carbon content was found to be a remarkable reduction in the N_2_/Ar group compared to those in other groups, indicating that the surface subjected to N_2_/Ar plasma irradiation was less susceptible to carbon contamination during the subsequent handling in the air. During the plasma treatment, the remnant organic compounds at the surface were removed by breaking a C–C bond [31]. The surface oxidation occurred with a formation of new functional groups (C–O and C=O bonds) [32], thereby enhancing the surface hydrophilicity [33]. The O 1s spectra, as shown in Figure 2B, consisted of a broad feature that can be resolved into three components: the lattice oxygen in ZrO_2_ (O_L_) at around 530.0 eV, the oxygen in acidic hydroxyl OH(a) group at around 531.5 eV, and the oxygen in basic hydroxyl OH(b) group at around 532.5 eV [29]. The surface hydroxyls may be formed by the dissociation of the moisture in the air at the specimen’s surface. Thus, the content of surface hydroxyls could increase when the surface provided the adsorption sites for H_2_O, such as oxygen-deficient defects. The relatively high O_L_-to-OH intensity ratio found in the N_2_/Ar group indicated that the energy for dissociative adsorption of H_2_O was relatively lower in the N_2_/Ar group than those in other groups, probably resulting from the surface oxynitride formation in the N_2_/Ar group (Figure 2B) through the partial nitridation of ZrO_2_ [34]. However, all plasma groups showed increased OH(b) groups (Figure 2D), suggesting the plasma treatments left the surface more defective toward dissociative H_2_O adsorption.

The XPS spectra of the Zr 3d spectra (Figure 2E) clearly revealed the two characteristic components of Zr 3d_3/2_ at 181.3 eV and of Zr 3d_5/2_ at 183.6 eV, which could be assigned to the zirconium in its Zr^4+^ state (ZrO_2_) [35]. In the Y 3d spectra (Figure 2F), two components of Y3d (Y3d_3/2_ and Y3d_5/2_) for the oxidized yttrium in its Y3^+^ state were identified. The pronounced Y3d exhibited in the N_2_/Ar group could be considered a result of grain refinement on the microstructure [17].

### 2.3. Crystallinity Changes by Irradiation of Various Gas Plasmas

The percentage of the phase compositions and the lattice parameters for each plasma group were calculated and the results are presented in Table 2. Before plasma irradiation (control), the zirconia phases observed were a tetragonal phase (t-ZrO_2_) as a major phase and a cubic phase (c-ZrO_2_). However, the formation of a metastable tetragonal phase (t’-ZrO_2_) was identified (up to 3 wt%) following the plasma treatments in all plasma groups. 

The powder XRD patterns and W-H plots of each specimen are shown in Figure 4. All the detected peaks corresponded to tetragonal and cubic phases, while no obvious monoclinic phase was observed. On analyzing the XRD peaks, a slight broadening of the tetragonal peak was seen in the N_2_/Ar group (Figure 4A,B), which was possible to deduce the contribution made by changes in the crystallite size and lattice strain [36,37]. By estimating from the slope and y-intercept of the W-H plot with Scherrer’s equation [36,37], strain and particle size were compared. Since the positive slope indicates the tensile strain [15], the development of a compressive strain could be estimated from the flatter slope in the N_2_/Ar group. A decrease in crystallite size, due to the lattice shrinkage and the occurrence of compressive strain in the N_2_/Ar group, could contribute to the peak broadening. The calculated crystallite sizes were 87.5 nm (control), 83.2 (He/O_2_), 65.1 (N_2_/Ar), 85.4 (N_2_), and 85.5 (Ar).

### 2.4. Surface Morphological Changes by Irradiation of Various Gas Plasmas

The magnified confocal images and SEM images of each plasma group are shown in Figure 5. The surface texture parameters (Sa, Sq, and Sv) measured from CLSM are given in Figure 6. With no significant morphological differences obtained, all specimens revealed relatively similar microstructures characterized by tetragonal symmetry integrated with large cubic crystals without relevant surface damages.

## 3. Discussion

This study investigated the effect of plasma gas compositions on the physicochemical surface characteristics of 3Y-TZP. A CAP discharge was carried out by using four different feed gases (He/O_2_ mixture, N_2_/Ar mixture, N_2_, and Ar) from an application distance of 10 mm for 60 s. According to the results of this study, the rate of decrease in water contact angle varied from 22.17% to 30.13%, while the rate of increase in surface free energy varied from 7.90% to 19.6%, depending on the plasma gas type. As previously reported [20], a lower contact angle indicated a more hydrophilic surface. In this study, a greater decrease in water contact angle was observed in the N_2_Ar group than in other plasma groups. This result was supported by the results of the XPS analysis; the N_2_Ar-specimen showed a lower carbon content and a higher oxygen content in comparison with the specimens subjected to other plasma gases. The dominant carbon species detected in XPS were generally associated with airborne carbon contaminants, which made the surface hydrophobic [38]. In the N_2_Ar group, C-C bonds in the hydrocarbon were broken or excited to metastable states by the collision cross-sections [39] between N and Ar, resulting in the formation of new functional groups that could generate a hydrophilic surface. The N_2_Ar-specimen had a greater amount of C-O species compared to other groups and this would be associated with the higher γ^p^ values in the N_2_Ar group. Feng et al. [40] reported that γ^p^ component in the surface energy influenced the cellular interaction more importantly compared to γ^d^ component. Considering the most noticeable changes in contact angle and γ^p^ component observed in the N_2_Ar group in this study, N_2_Ar plasma could lead to higher bioactivity of the zirconia specimen compared to other plasma groups.

Partially stabilized zirconia, suitable for dental applications, is obtained by the addition of lower-valence oxide, such as 3 mol% (5.2 wt%) yttria. The oxygen vacancies, which are introduced to compensate for the charge imbalance, are responsible for the high ionic conductivity of the zirconia material [41]. In this study, the dissociation of water, which would create the OH(b) groups per vacancy, occurred in all plasma groups, resulting in more positive surface charges [40]. A greater increase in OH(b) was observed in the N_2_ group than in other groups. A previous study reported that the OH(b) could play a more significant role than the OH(a) to enhance bioactivity of the substrate because the primary protein can be easily attracted to positively charged surfaces [16].

Unlike other plasma gases, N_2_Ar plasma led to the formation of nitrate (NO_3_^−^) species on the zirconia surface, according to the results of the XPS analysis in this study. The nitrate anion can produce a highly reactive nitrate radical (^•^NO_3_), which can react with organic compounds due to high diffusivity into non-polar solvents [42]. Thus, this may enhance the bioactivity or bonding efficiency of 3Y-TZP. The ubiquitous formation of zirconium oxynitride (ZrO_x_N_y_) investigated in all plasma groups can be proposed as being associated with high oxygen concentration in the superficial regions of zirconia specimens. It has been reported that the zirconium oxynitride could be formed when the N concentration did not reach a critical value to transform it into the ZrN structure [29]. The plasma nitriding of zirconia could produce ZrN structures, which were characterized by high hardness, even harder than 3Y-TZP, high resistance to wear or corrosion, and high thermal stability [29]. Morisaki et al. [30] revealed that the nitridation of zirconia was generated by a replacement of oxides with nitrides, and the lattice structure of the zirconium oxynitride was slightly deformed from the ideal cubic structure. The results of this study are consistent with those obtained in Morisaki et al.’s study [30]. The Rietveld refinement showed a decrease in the cubic phase content and an increase in the metastable tetragonal (t’) phase content in both N_2_Ar and N_2_ groups, suggesting the possible effects of the plasma nitriding on the lattice distortion [43]. The t’ phase was formed in all plasma groups, which can be attributed to the induced oxygen atom displacements in the zirconia crystal structures [43] during the plasma irradiation. Considering the XRD peak broadening and the flatter slope from the W-H analysis obtained in the N_2_Ar group, the plasma nitriding of 3Y-TZP would induce lattice strain through the crystal deformation [44]. This may play an important role in the enhancement of mechanical properties by increasing damage resistance [45]. A similar result was reported by Abbas et al. [46], who found the broadened XRD peaks in silicon samples subjected to N_2_Ar plasma. They also concluded that the changes in crystallinity, crystallite sizes, residual stress, and surface morphology of Si samples were related to Ar concentration in N_2_Ar plasma.

As mentioned, an admixture of molecular gas to a chemically inert noble gas, such as helium, neon, argon, etc., can change the plasma discharge kinetics, showing better biologic performance [23]. For N_2_ plasma, its dissociation is very difficult because of the extremely strong triple bond between two N atoms [47]. A previous study revealed that the addition of argon to the N_2_ plasma resulted in the enhanced generation of active species through Pennig excitation and ionization, depending on the Ar concentration in N_2_Ar plasma [46]. Bravo et al. [47] reported that the metastable nitrogen molecules were produced utilizing a plasma gas mixture containing N_2_ and Ar. Dyatko et al. [23] found that even a small amount of nitrogen addition (1%) to argon was responsible for a sharp decrease in the discharge voltage in N_2_Ar plasma as compared to that in pure Ar gas. This behavior of the N_2_Ar discharge involved the electron collision cross-section. This study used N_2_Ar plasma containing 10% nitrogen, which was reported to be the most frequently used concentration for the application to surface treatments [47]. During the N_2_Ar plasma irradiation on the zirconia surface, nitrogen atoms can be replaced by oxygen atoms and thus the zirconia surface tends to have more negative charges. The most suitable nitrogen concentration in N_2_Ar plasma should be determined for the production of bioactive zirconia surface in further studies. Although helium plasma can easily give rise to a stable glow discharge with the addition of an active gas such as O_2_, N_2_, or CF_4_, reactive oxygen species cannot easily reach the target material because helium is considerably lighter than air [48]. Contrary to helium, argon is denser than air and thus, the excited atomic oxygen can be easily transferred to the substrate [48]. According to the result of this study (Figure 3A), an increase in the reactive oxygen species upon adding nitrogen to argon was observed.

In this study, different plasma gas species produced different physicochemical surface interactions with 3Y-TZP, although the plasma treatments did not affect the surface morphology of zirconia as shown by SEM and CLSM analysis. Therefore, the null hypothesis was rejected. From a biomaterials perspective, biologic responses and, hence, the clinical function of 3Y-TZP can be enhanced by controlling the plasma gas type and the working gas composition. However, the interpretation of the present results should be done with caution since an aging effect such as a time-dependent hydrophobic recovery can occur at the zirconia surface, which would act as a limiting factor on the efficiency of the plasma treatments [17]. Further investigations are required to identify the time scale for the hydrophobic recovery of the zirconia surface subjected to the plasma treatment with various gas species to determine the effective treatment time of plasma discharge to improve the bioactivity of zirconia surfaces.

## 4. Materials and Methods

### 4.1. Specimen Preparation and Plasma Surface Treatment

The 3Y-TZP (KATANA ML, Kuraray Noritake Dental, Osaka, Japan) sintered at 1500 °C for 2 h in air was used in this study. A total of 140 plate-shaped specimens (10.0 mm × 10.0 mm × 1.0 mm) were prepared and polished to a uniform finish with an 800 grit SiC paper. After ultrasonic cleaning for 20 min, the specimens were subjected to plasma irradiation at room temperature using a cold atmospheric pressure DBD plasma generator (PR-ATO-001, ICD Co., Anseong, Gyeonggi-do, Korea). The plasma was vertically applied to the specimen’s surface from a distance of 10 mm [39] for 60 s. The schematic of the device is shown in Figure 7. All specimens were randomly allocated into five groups (*n* = 28), with four of them treated by plasmas with four different gases and one group untreated (control). As the feed gases, Ar, N_2_, N_2_/Ar mixture (10% nitrogen/90% argon) [7], and He/O_2_ mixture (15% helium/85% oxygen) [8] were selected. The input voltage was fixed at 5 kV with a high voltage transformer, and the operation frequency was set to 25 kHz using a digital oscilloscope (MSO4032, Tektronix, Beaverton, OR, USA). A mass flow controller maintained a constant gas flow rate of 10 standard liters per minute (slm).

### 4.2. Surface Contact Angle and Surface Free Energy

The surface wettability of the specimen was characterized using a contact angle meter (Phoenix 300 Touch, S.E.O., Suwon, Gyeonggi-do, Korea). Contact angle measurements were performed by a sessile drop technique at room temperature and 60% relative humidity using two different test liquids, distilled water (*n* = 10) and non-polar diiodomethane (*n* = 10). All measurements were performed at the center of the specimen.

The surface free energy was calculated by measuring the contact angles of two test liquids according to the Owens–Wendt equation [49]. Total surface free energy (γ^total^), including the dispersion component(γ^d^) and polar component (γ^p^), was calculated. 

### 4.3. X-ray Photoelectron Spectroscopy (XPS) Measurement

The elemental composition of the specimen of each group was performed by using XPS (K-alpha, Thermo Fisher Scientific Inc., Waltham, MA, USA) with a monochromatic Al Kα X-ray source (1486.6 eV) at 12 kV and 3 mA (*n* = 1). Data acquisition and the processing of core-level spectra were performed using a software (Thermo Avantage v5.980, Thermo Fisher Scientific Inc., Waltham, MA, USA). All XPS spectra were calibrated with the C 1s peak at 284.6 eV [26].

### 4.4. X-ray Diffraction (XRD) and Rietveld Analysis

One specimen from each plasma group was submitted to determine the crystal structures and phase transformations. The X-ray powder diffraction (XRD) pattern was taken at room temperature in a DMAX-2200PC X-ray diffractometer (Rigaku, Tokyo, Japan) using monochromatic CuKα1 radiation (*λ* = 1.5406 Å). The data were collected in the 2θ range of 20–90 with a step size of 0.02 and a step time of 4 s/step. The quantitative phase analysis was obtained by Rietveld refinement implemented in the Fullprof program [15]. Especially, Williamson–Hall (W–H) analysis was used to determine the changes in the crystallite size and the lattice strain induced by the plasma treatment using the formula as given below [50]:(1)$$\beta_{\mathrm{hkl}}\;\cos\theta =\frac{K\lambda }{D}+4ɛ\;\sin\theta$$
where β is the integral breadth or the full width at half maxima, *D* is the crystallite size, *K* is a shape factor (0.9), and ɛ is the strain.

### 4.5. Surface Topography

Analyses of three-dimensional (3-D) surface characteristics were performed using a confocal laser scanning microscopy (CLSM; LEXT OLS3000, Olympus, Tokyo, Japan) at 50× magnification in a 256 × 192 μm^2^ area (*n* = 5). The surface texture parameters, in particular the arithmetical mean height, Sa; the root mean square height, Sq; and the maximum pit height, Sv, were calculated in accordance with ISO 25,178 [51]. The surface analysis was independently carried out in 2 points at the center, and a total of 10 measurements was obtained for each plasma group.

The surface microstructures of the specimens were evaluated using a scanning electron microscope (SEM; JSM-7800F Prime, JEOL, Tokyo, Japan) at an accelerating voltage of 5.0 kV and a working distance (WD) of 6.0 mm at 3000×, 10,000×, and 30,000× magnifications (*n* = 1).

### 4.6. Statistical Analysis

The statistical significance of the data was assessed by a one-way analysis of variance (ANOVA) with Tukey’s honesty significant difference (HSD) post hoc test at α = 0.05. All analyses were performed using statistical software (IBM SPSS Statistics, v25.0, IBM Corp., Chicago, IL, USA).

## 5. Conclusions

We have investigated the effect of plasma gas types on surface physicochemistry and surface topography of 3Y-TZP by contact angle, XPS, XRD, CLSM, and SEM analysis. With the limitation of this in vitro study, the following conclusions can be drawn:Plasma processing of 3Y-TZP with He/O_2_ mixture, N_2_/Ar mixture, N_2_, and Ar decreased the contact angle and increased the surface energy without changing its surface topography. In particular, the highest value of the polar component in the surface energy was obtained in the N_2_/Ar group, which was probably related to the high interaction energy of the zirconia surface.XPS revealed that a reduction in the surface C content along with an increase in O content was pronounced in the case of N_2_/Ar compared to others, which was responsible for high hydrophilicity of the surface. XRD showed that the changes in crystallite size and microstrain due to oxygen atom displacements were observed in the N_2_/Ar group.Although further studies are required, promising findings obtained in this study show that N_2_/Ar plasma treatment may contribute to enhancing the anti-microbial properties, the osseointegration capability, and the adhesion performance of 3Y-TZP by controlling the plasma-generated nitrogen functionalities.

## Figures and Tables

**Figure 1 ijms-23-03007-f001:**
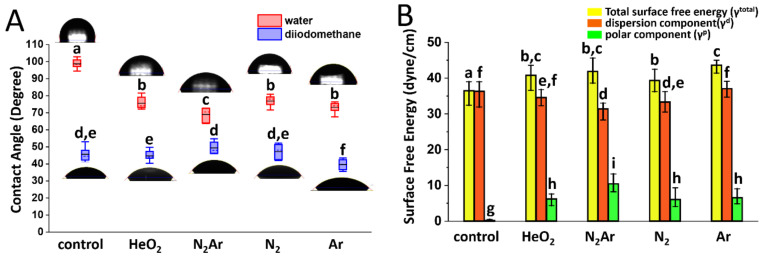
(**A**) Contact angles of water and diiodomethane; box plot showing the mean (black solid horizontal line), median (dashed horizontal line), interquartile range (box), and total range (whiskers) of the data set. (**B**) Values of the total surface free energy (γ^total^), dispersion component (γ^d^), and polar component (γ^p^) in the zirconia specimens subjected to each plasma treatment. An identical letter within each value indicates no significant difference between each type of plasma gas (*p* > 0.05). The error bar represents the standard deviation.

**Figure 2 ijms-23-03007-f002:**
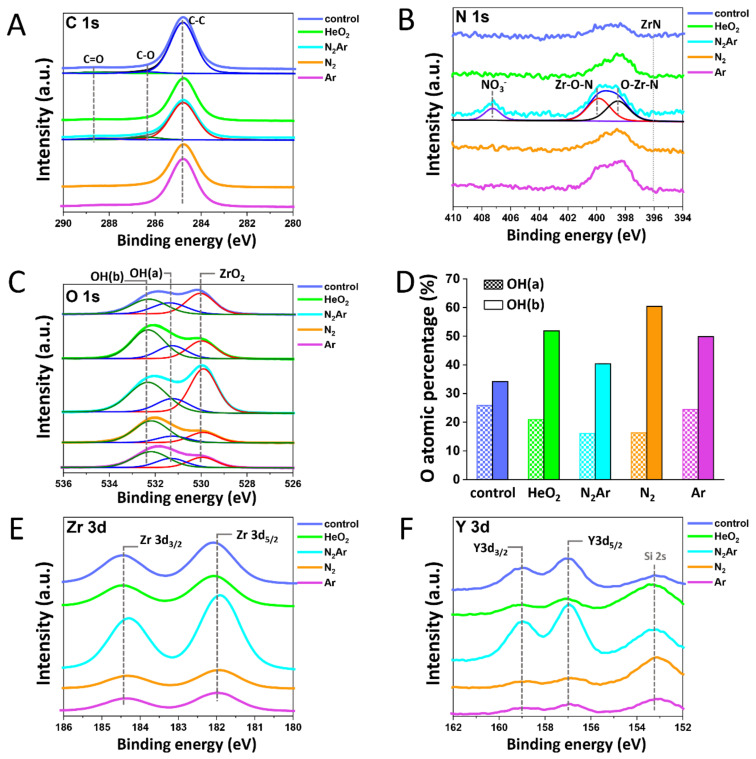
(**A**) C 1s; the carbon content was found to be a remarkable reduction in the N_2_/Ar group compared to those in other groups. (**B**) N 1s; a new component appeared at a binding energy of 406.5 eV in the N_2_/Ar group, which was associated with the presence of nitrate (NO_3_^−^) species. (**C**) O 1s XPS spectra; (**D**) Percentage areas of acidic hydroxyl OH(a) and basic hydroxyl OH(b) in O 1s spectra; (**E**) Zr 3d; and (**F**) Y 3d XPS spectra on outermost surfaces of zirconia with different plasma gases.

**Figure 3 ijms-23-03007-f003:**
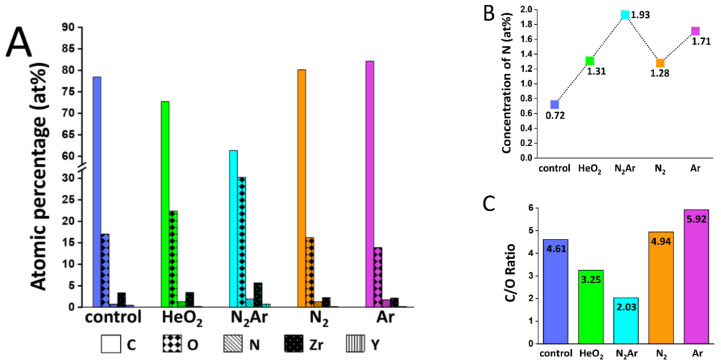
(**A**) Atomic percentage (at%) for each element detected in each plasma group; (**B**) Concentration of N in each plasma group; and (**C**) Carbon/oxygen ratio in each plasma group.

**Figure 4 ijms-23-03007-f004:**
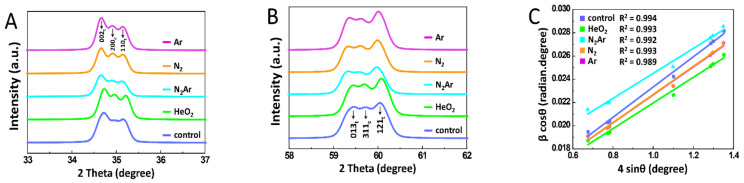
X-ray diffraction (XRD) patterns of zirconia specimens with each plasma gas (**A**) in the ranges of 2θ = 33–37°, (**B**) of 2θ = 58–62°, (**C**) Williamson-Hall (W-H) plot of ß cos θ against 4 sin θ of the tetragonal phases calculated from XRD spectra.

**Figure 5 ijms-23-03007-f005:**
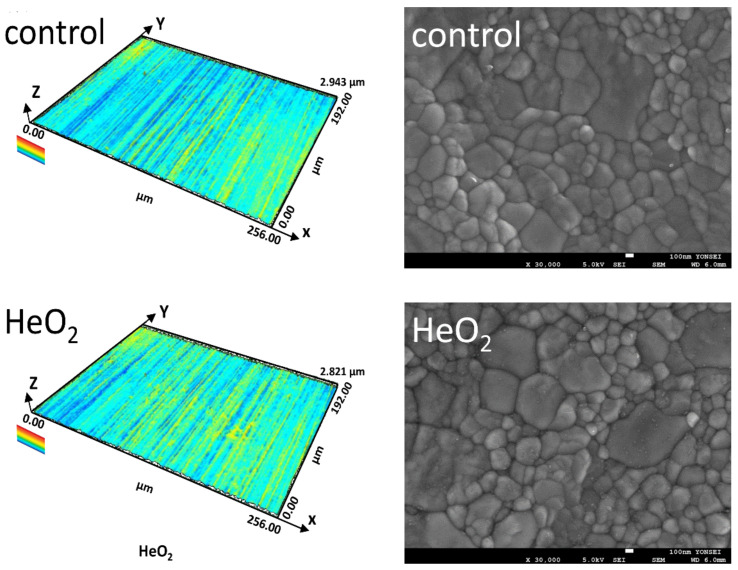
Representative three-dimensional images obtained by confocal laser-scanning microscopy (**left**) and scanning electron microscopic images at 30,000× magnification (**right**) of each group.

**Figure 6 ijms-23-03007-f006:**
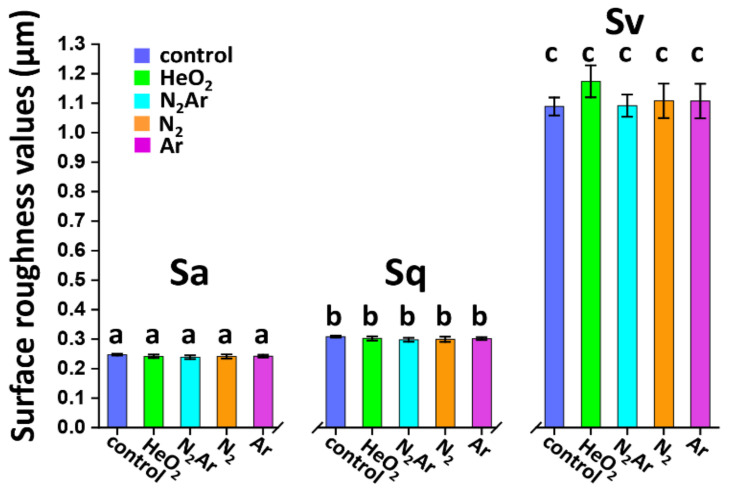
The surface texture parameters (Sa, Sq, and Sv) of each group. The identical letter shows no significant difference on Sa, Sq, and Sv values (*p* > 0.05).

**Figure 7 ijms-23-03007-f007:**
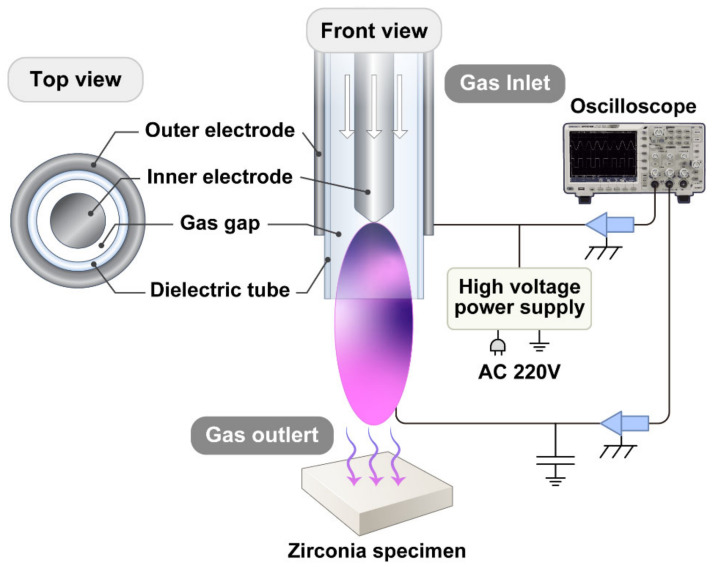
Schematic diagram of experimental setup for plasma surface treatments.

**Table 1 ijms-23-03007-t001:** Contact angles for water and diiodomethane on the zirconia surfaces of each plasma group.

Plasma Group	Contact Angle (°)	
	Water	Diiodomethane
control	98.75 ± 2.70 ^a^	45.66 ± 4.30 ^d,e^
HeO_2_	75.59 ± 3.38 ^b^	44.72 ± 3.16 ^e^
N_2_Ar	69.00 ± 3.98 ^c^	49.39 ± 3.33 ^d^
N_2_	76.86 ± 3.30 ^b^	47.21 ± 4.14 ^d,e^
Ar	73.22 ± 3.00 ^b^	39.60 ± 3.19 ^f^

Means with the same superscript letter in each column are not significantly different from each other based on Tukey’s honest significant difference post hoc test (*p* > 0.05).

**Table 2 ijms-23-03007-t002:** Rietveld analysis results for the phase compositions and the lattice parameters in each plasma group.

Plasma Group	Phase	Amount (wt%)	Lattice Parameters		
			a = b (Å)	c (Å)	*c*/*a* Ratio
control	t	62(2)	3.6069(2)	5.1777(4)	1.0151
	c	38(2)	5.1382(3)	5.1382(3)	
HeO_2_	t	59(2)	3.6070(2)	5.1788(4)	1.0152
	t’	2(1)	3.625(2)	5.173(5)	1.0091
	c	39 (1)	5.1383(3)	5.1383(3)	
N_2_Ar	t	67(2)	3.6098(2)	5.1808(4)	1.0148
	t’	3(1)	3.626(2)	5.175(5)	1.0092
	c	30(1)	5.1423(3)	5.1423(3)	
N_2_	t	66(2)	3.6087(2)	5.1804(4)	1.0151
	t’	3(1)	3.626(1)	5.172(3)	1.0086
	c	31(1)	5.1407(3)	5.1407(3)	
Ar	t	60(2)	3.6075(2)	5.1779(4)	1.0149
	t’	2(1)	3.625(2)	5.173(6)	1.0091
	c	38(1)	5.1390(3)	5.1390(3)	

t: tetragonal zirconia (space group *P4_2_/nmcS*); t’: metastable tetragonal zirconia (space group *P4_2_/nmcZ*); c: cubic zirconia (space group *Fm*$\bar{3}$*m)*; Values in parentheses correspond to the estimated standard deviation in the least significant figure to the left. *c*/*a* ratio = c (Å)/$\sqrt{2}$ a (Å).

## Data Availability

The data presented in this study are available on request from the corresponding author.

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
