# Peer review of "Effect of the Plasma Gas Type on the Surface Characteristics of 3Y-TZP Ceramic"

_ijms, 2022, doi:10.3390/ijms23063007_

Round 1
Reviewer 1 Report
See attached document.

Author Response
Dear Reviewer,
We, the authors, highly appreciate the detailed valuable comments on this manuscript.
The revision was listed below the comments and recommendations one by one.
We also highlighted the revised parts in red color for your convenience in the revised paper.
===================================================
We specified the type of zirconia, 3Y-TZP, used in this study instead of ‘dental zirconia’ in the revised manuscript. We also highlighted the revised parts in red color for your convenience in the revised paper. Thank you for the recommendation.
Title: Effect of the Plasma Gas Type on the Surface Characteristics of 3Y-TZP ceramic
Abstract: 3 mol% yttria-stabilized zirconia (3Y-TZP)~
Introduction: Zirconia ceramics stabilized with 3 mol% of yttria (3Y-TZP) have been ~
Discussion: 3Y-TZP~
Conclusion: 3Y-TZP~

Reviewer 2 Report
This study concerns an influence of the plasma gas composition on physicochemical properties of dental zirconia surface. Fout types of the plasma gas were applied to the specimen’s surface and compared with control specimen (without treatment). Several experimental methods such as a contact angle, XPS, XRD, CLSM and SEM were used to characterize surface properties after plasma gas treatment. The study provided a worth of experimental findings that are valuable and contribute to the knowledge base. However, the paper needs some improvement and explanation of unclear issues. Below some specific questions and comments are listed.
1. The structure of the article should be changed. Chapter no 4: Materials and methods should be given before chapter no 2: Results.
2. Figures’ captions (e.g., Fig. 1, 5, 6) contain a copied description of the test results, which can be found in the manuscript. I think that discussion of the experimental results should only be in the manuscript body instead of figure caption.
3. The results given in the figure 1(A) are illegible. The figure shows the trend, but the values of contact angle and standard deviations are difficult to compare. Maybe the figure should be supplemented with a table.
4. Fig. 2, description of (D) should be moved before (E).
5. Lack of reference in the text to the figure 3 and description of the results.
6. In the conclusions, in the firs statement, the Authors mentioned about “mechanical changes”. This statement is too general and does not fully refer to the results presented in the study. Conducted analysis with the use of W-H plot and Scherrer’s equation allowed to discuss character of lattice strain, but further analysis of stress or modulus of elasticity was not conducted.
Author Response
Dear Reviewer,
We, the authors, highly appreciate the detailed valuable comments on this manuscript.
The revision was listed below the comments and recommendations one by one.
We also highlighted the revised parts in red color for your convenience in the revised paper.
===================================================
Point 1: The structure of the article should be changed. Chapter no 4: Materials and methods should be given before chapter no 2: Results.
Response 1:
We changed the structure of the article in the revised manuscript.
Point 2: Figures’ captions (e.g., Fig. 1, 5, 6) contain a copied description of the test results, which can be found in the manuscript. I think that discussion of the experimental results should only be in the manuscript body instead of figure caption.
Response 2: We removed a copied description of the test results in figure captions in the revised manuscript.
Figure 2. (A) Contact angles of water and diiodomethane; box plot showing the mean (black solid horizontal line), median (dashed horizontal line), interquartile range (box), and total range (whiskers) of the data set. (B) Values of the total surface free energy (γtotal), dispersion component (γd), and polar component (γp) in the zirconia specimens subjected to each plasma treatment. Identical letter within each value indicates no significant difference between each type of plasma gas (p > 0.05). The error bar represents the standard deviation.
Figure 3. (A) C 1s; the carbon content was found to be a remarkable reduction in the N2/Ar group compared to those in other groups. (B) N 1s; a new component appeared at a binding energy of 406.5 eV in the N2/Ar group, which was associated with the presence of nitrate (NO3-) species. (C) O 1s XPS spectra; (D) Percentage areas of acidic hydroxyl OH(a) and basic hydroxyl OH(b) in O 1s spectra; (E) Zr 3d, and (F) Y 3d XPS spectra on outermost surfaces of zirconia with different plasma gases.
Figure 4. (A) Atomic percentage (at%) for each element detected in each plasma group; (B) Concentration of N in each plasma group; (C) Carbon/oxygen ratio in each plasma group.
Figure 5. X-ray diffraction (XRD) patterns of zirconia specimens with each plasma gas (A) in the ranges of 2θ = 33–37°, (B) of 2θ = 58–62°, (C) Williamson-Hall (W-H) plot of ß cos θ against 4 sin θ of the tetragonal phases calculated from XRD spectra.
Figure 6. Representative three-dimensional images obtained by confocal laser scanning microscopy (left) and scanning electron microscopic images at 30,000× magnification (right) of each group.
Figure 7. The surface texture parameters (Sa, Sq, and Sv) of each group. Identical letter shows no significant difference on each Sa, Sq, and Sv values (p > 0.05).
Point 3: The results given in the figure 1(A) are illegible. The figure shows the trend, but the values of contact angle and standard deviations are difficult to compare. Maybe the figure should be supplemented with a table.
Response 3: We added a table (Table 1) in the revised manuscript. Thank you for the recommendation.
Table 1. Contact angles for water and diiodomethane on the zirconia surfaces of each plasma group.
|
Plasma group |
Contact angle (°) |
|
|
Water |
Diiodomethane |
|
|
control |
98.75 ± 2.70a |
45.66 ± 4.30d,e |
|
HeO2 |
75.59 ± 3.38b |
44.72 ± 3.16e |
|
N2Ar |
69.00 ± 3.98c |
49.39 ± 3.33d |
|
N2 |
76.86 ± 3.30b |
47.21 ± 4.14d,e |
|
Ar |
73.22 ± 3.00b |
39.60 ± 3.19f |
Means with the same superscript letter in each column are not significantly different from each other based on Tukey’s honestly significant difference post hoc test (p > 0.05).
Point 4: Fig. 2, description of (D) should be moved before (E).
Response 4: We moved the description of (D) before (E) in Figure 3 (Figure 2 was changed to Figure 3 according to the structural change of the manuscript.)
Figure 3. (A) C 1s; the carbon content was found to be a remarkable reduction in the N2/Ar group compared to those in other groups. (B) N 1s; a new component appeared at a binding energy of 406.5 eV in the N2/Ar group, which was associated with the presence of nitrate (NO3-) species. (C) O 1s XPS spectra; (D) Percentage areas of acidic hydroxyl OH(a) and basic hydroxyl OH(b) in O 1s spectra; (E) Zr 3d, and (F) Y 3d XPS spectra on outermost surfaces of zirconia with different plasma gases.
Point 5: Lack of reference in the text to the figure 3 and description of the results.
Response 5: We added the reference for Figure 4 (Figure 3 was changed to Figure 4 according to the structural change of the manuscript) in the text:
Figure 3 shows the XPS C 1s, O 1s, N 1s, Y 3d, and Zr 3d core-level spectra and Figure 4 shows the atomic percentages (at%) of these elements determined by XPS and the carbon/oxygen ratio in all groups. The nitrogen content on the zirconia surface increased after the plasma treatment in all plasma groups, but only to a few percent range (1–2%) as illustrated in Figures 4A and 4B. An increase in oxygen content and a decrease in carbon content were found in both N2/Ar and He/O2 groups, with the largest decrease in C content and the largest increase in O content in the N2/Ar group (Figure 4A). As can be seen in Figure 4C, the lowest value of C/O ratio was observed in the N2/Ar group which would be related to the largest increase in the surface hydrophilicity in the N2/Ar group, possibly by the production of a high level of oxygen~
Point 6: In the conclusions, in the firs statement, the Authors mentioned about “mechanical changes”. This statement is too general and does not fully refer to the results presented in the study. Conducted analysis with the use of W-H plot and Scherrer’s equation allowed to discuss character of lattice strain, but further analysis of stress or modulus of elasticity was not conducted.
Response 6: We changed the first statement in the conclusions: We have investigated the effect of plasma gas types on the surface physicochemistry and surface topography of 3Y-TZP by contact angle~
